# Anomalous Dirac point transport due to extended defects in bilayer graphene

Sam Shallcross[1], Sangeeta Sharma[2] & Heiko B. Weber [3,4]

Charge transport at the Dirac point in bilayer graphene exhibits two dramatically different transport states, insulating and metallic, that occur in apparently otherwise indistinguishable experimental samples. We demonstrate that the existence of these two transport states has its origin in an interplay between evanescent modes, that dominate charge transport near the Dirac point, and disordered configurations of extended defects in the form of partial dislocations. In a large ensemble of bilayer systems with randomly positioned partial dislocations, the distribution of conductivities is found to be strongly peaked at both the insulating and metallic limits. We argue that this distribution form, that occurs only at the Dirac point, lies at the heart of the observation of both metallic and insulating states in bilayer graphene.

[1] Lehrstuhl für Theoretische Festkörperphysik, Staudstr. 7-B2, 91058 Erlangen, Germany. [2] Max-Planck-Institut für Mikrostrukturphysik Weinberg 2, D-06120 Halle, Germany. [3] Lehrstuhl für Angewandte Physik, Staudtstr. 7, 91058 Erlangen, Germany. [4] Interdisziplinäres Zentrum für Molekulare Materialien, Friedrich-Alexander-Universität Erlangen-Nürnberg, Erlangen 91058, Germany. Correspondence and requests for materials should be addressed to S. Shallcross (email: sam.shallcross@fau.de)

Transport at the Dirac point in structurally perfect bilayer graphene is expected to exhibit a minimal metallic conductivity, very similar to that found in single layer graphene[1–3]. Contradicting this expectation, recent transport experiments on ultra-clean bilayer graphene report the existence, near charge neutrality, of an insulating phase. Curiously, this occurs only in about 50% of an apparently identical set of ultra-clean high-mobility samples, with the remainder showing the expected minimal metallic conductivity[4–8]. Two different explanations for the existence of the insulating state have been proposed: that the eight-fold degeneracy of the bilayer electron fluid at the Dirac point is responsible for an interaction-driven symmetry breaking to an insulating phase[4–7] or, alternatively, that the insulating state is a manifestation of charge blocking by partial dislocations[8], a structural defect recently observed in many bilayer graphene samples[9–12].

In both these models the occurrence in experiment of metallic and insulating states with approximately equal probability is difficult to explain. In the interaction model we have the difficulty of apparently identical ultra-clean samples resulting only sometimes in the transition to an insulating phase; similarly, in the blocking model the metallic state requires for its existence samples that are entirely free of partial dislocations, a situation that would appear unlikely.

Here we resolve this conundrum by identifying an unusual transport physics that arises from the interplay of extended defects and evanescent Dirac point transport in bilayer graphene. We find that some positions of partial dislocations entirely block transport while others appear not to impede charge transport at all. In particular, in an ensemble of systems with randomly placed partials, the distribution function of conductivities has pronounced—and approximately equal—maxima at both the insulating and the metallic limits. This behaviour, which sharply deviates from the well-known paradigm that disorder should generally act to suppress transport, we argue that this underpins the observation of both insulating and metallic transport states in experiment.

## Results

**Model.** The microscopic origin of partial dislocations arises from the fact that Bernal stacking may be achieved in two equivalent ways, typically referred to as AB and AC stacking, as shown in Fig. 1a. While evidently equivalent in a bilayer of infinite extent, their difference becomes significant if both types coexist in the same sample. Lattice continuity then requires that domains of different stacking order be connected by partial dislocations, localised regions in space within which the transition between stacking types occurs. Three such partial dislocations are possible in bilayer graphene[9–12], described by one of the three partial Burgers vectors: $\mathbf{d}_1 = a\left[\frac{1}{2}, \frac{1}{2\sqrt{3}}\right]$, $\mathbf{d}_2 = a\left[-\frac{1}{2}, \frac{1}{2\sqrt{3}}\right]$ and $\mathbf{d}_3 = a\left[0, -\frac{1}{\sqrt{3}}\right]$ (where $a$ is the graphene lattice parameter, and $\mathbf{d}_i$ the nearest neighbour vectors of the honeycomb lattice). Note that these stacking fault partial dislocations are very different in character from the dislocations found in single-layer

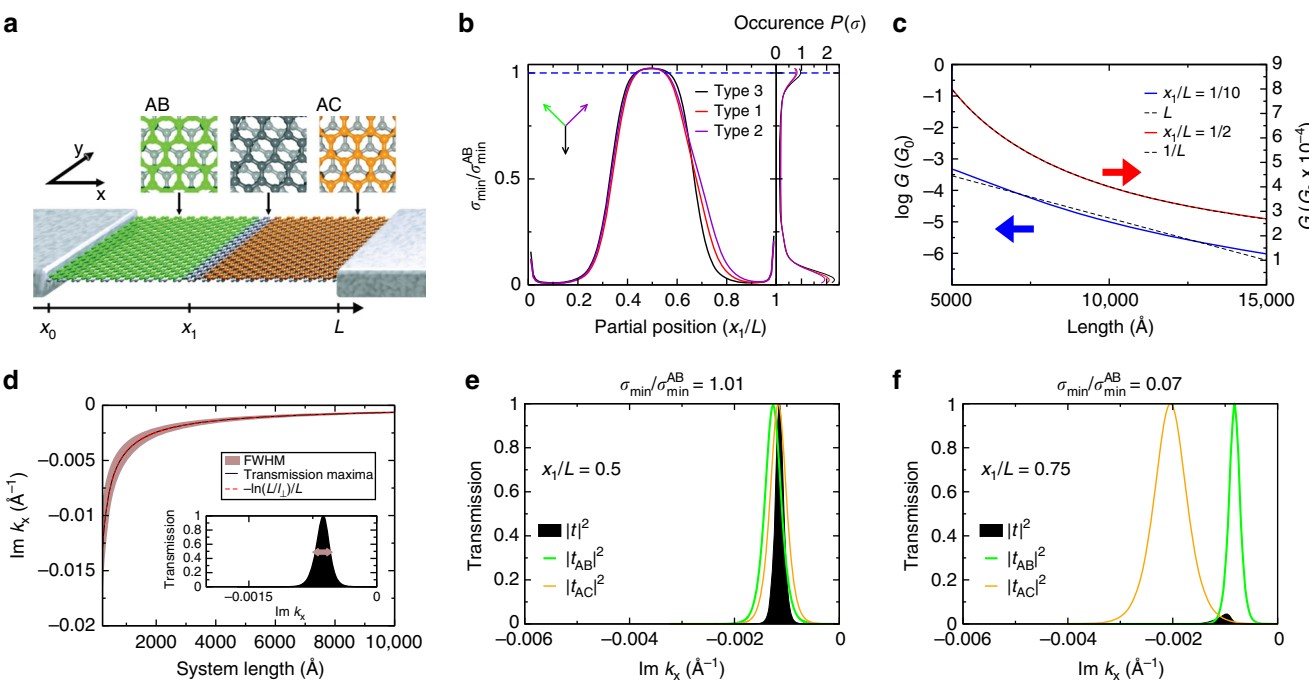

**Fig. 1** Resonant and blocked transport states for a single partial dislocation in bilayer graphene. **a** Schematic illustration of a bilayer graphene ribbon consisting of two domains of structurally perfect Bernal stacking connected by a single partial dislocation, with the *grey area* depicting the finite non-Bernal transition region and the *upper panels* displaying the structure of the local AB, non-Bernal, and AC stacking geometries. **b** Conductivity $\sigma$ at the Dirac point as a function of the partial position $x_1/L$, ranging from minimal metallic conductivity at $x_1/L = 1/2$ to insulating behaviour at $x_1/L = 2/10$. Calculations are shown for three different partial Burgers vectors of the partial dislocation. The *right hand plot* shows the probability distribution function of conductivities, indicating the likelihood of occurrence for a given transport state. **c** Dependence of the conductance $G$ on sample length for the metallic ($x_1/L = 1/2$) and insulating ($x_1/L = 2/10$) partial positions, showing $1/L$ and exponential dependence, respectively. **d** Transport through a single terrace, showing the transmission function for $L = 1\,\mu\text{m}$ (inset), and (main panel) the maximum and full width half maximum (*FWHM*) of the transmission resonance as a function of length. **e**, **f** Transmission probability of charge carriers through the full system ($|t|^2$), and through the individual AB ($|t_{AB}|^2$) and AC ($|t_{AC}|^2$) terraces separately connected to leads, for a conducting and insulating partial position. Evidently underpinning the conducting state is overlap of the individual terrace transmission resonance peaks **e**, with transport blocking occurring when these are well separated (**f**)

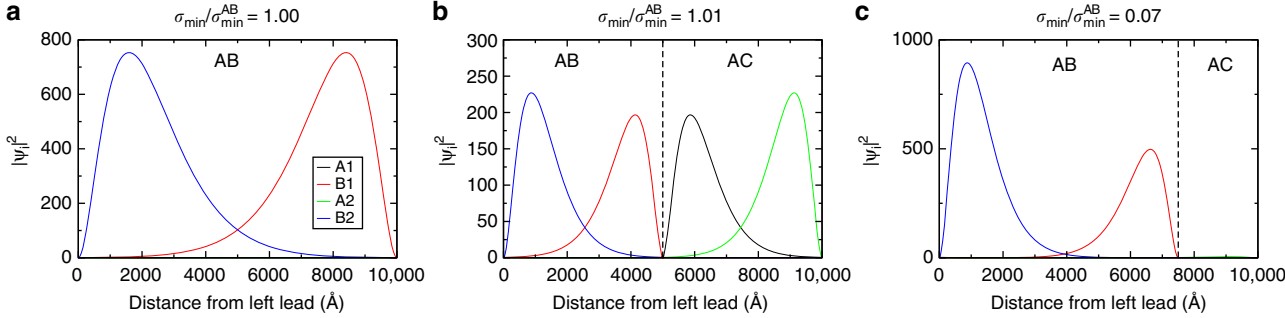

**Fig. 2** Transport states $|\Psi_i(x)|^2$ at the Dirac point. The *vertical lines* represent the positions of the partial dislocations, which can be seen to determine the basic structure of the wavefunction, while the *line colour* refers to the sublattice projection as indicated by the legend in **a**. Shown are: **a** the dislocation free bilayer, **b** conducting and **c** insulating states in a single partial configuration. Significant amplitude of $|\Psi_i(x)|^2$ occurs only on the non-bonding sublattices, which are B1 and B2 for the AB stacked geometry, and A1 and A2 for the AC stacked geometry

graphene[13, 14]. Indeed, as they require for their existence only local strain within the constituent layers of the bilayer, they may occur between graphene sheets that are essentially structurally perfect. The weak interlayer bonding of bilayer graphene then implies a rather low energy cost of stacking fault partial dislocations and they are, for example, found to form a dense network in epitaxial graphene grown on SiC (0001)[9, 10], where they arise due to strain induced by the lattice mismatch between graphene and the SiC substrate.

In order to understand the impact of such stacking defects on charge transport, we will consider a model of partial dislocations that are parallel with the left and right electrodes contacting the bilayer sample (Fig. 1a). This is the simplest conceivable structural model, and has advantages both in tractability (as it results in an effectively one dimensional model) as well as, most importantly for our purpose here, lending itself to a transparent analysis of the role of structural disorder. We note that while the imposition of a straight partial geometry might appear a rather restrictive condition, substantially simplifying mode matching at the partials, the difference between straight and non-straight partials was found not to be qualitatively significant in ref. [8]. We will return to this point subsequently. Even with this simplifying assumption, however, modelling the transport experiments remains a challenge due to the presence of two very different length scales: the partial dislocation which occurs on a scale of $\approx 5$ nm, and the length of the suspended bilayer graphene sample which is typically of the order of 1 µm. This is therefore a multi-scale problem, and evidently precludes the use of an atomistic approach such as the tight-binding method. To circumvent this difficulty we will make use of a recently developed effective Hamiltonian theory capable of treating large-scale structural deformations in low-dimensional systems[15], and shown recently to provide a very good description of the electronic structure both for partial dislocations[11, 15] as well as twist faults in bilayer graphene[16–18]. In this scheme the individual layers of the bilayer are modelled by the Dirac–Weyl Hamiltonian $H_0 = v_F \boldsymbol{\sigma}.\mathbf{p}$ (with gauge fields arising from the strain fields of the partial included), while the spatially dependent stacking order is represented by an interlayer stacking field that, importantly, treats all stacking types on an equal footing (i.e., the approach is non-perturbative). The transport problem is then addressed within the framework of Landauer theory with the leads modelled as highly doped bilayer graphene[1, 2]. Further details may be found in the Methods section.

**Transport through a single partial dislocation.** We first apply this scheme to the single partial configuration of Fig. 1a, finding a large range of conductivities as a function of partial position

(Fig. 1b). These include both a slightly resonantly enhanced minimal conductivity when the partial is at the high symmetry central position, as well as an essentially insulating behaviour when the partial moves out of this region towards one of the leads. The conductance $G$ shows the expected $L^{-1}$ (minimal metallic) or $\exp(-L/L_0)$ (insulating) dependence on the system length for the cases $x_1/L = 1/2$ and $x_1/L = 2/10$, respectively (Fig. 1c). The resonant metallic state is, in fact, very similar to that found for the structurally perfect AB bilayer by Snyman et al.[2], and indeed the Fano factor is one-third, exactly the value found for structurally perfect bilayer graphene. Strikingly, this simple model already shows the preferential occurrence of two transport states, as may be seen from the probability density function $P(\sigma)$ (see Fig. 1b, $\int_{\sigma_1}^{\sigma_2} d\sigma P(\sigma)$ is the probability of finding a conductivity $\sigma_1 < \sigma < \sigma_2$ given a random choice of partial position). As may be seen, this function has pronounced peaks at both the insulating and minimally metallic transport states. These results, it should be stressed, are rather independent of the lead structure, the strain state of the partial, and indeed the partial Burgers vector—as may be seen in Fig. 1b, a similar $\sigma(x_1)$ dependence is found for all three possible partial Burgers vectors of bilayer graphene (the direction of the partial Burgers vectors for each dislocation type is shown in the inset of Fig. 1b).

The origin of this unusual two-state transport is, as we now show, driven by the transmission properties of individual bilayer terraces at the Dirac point. Assuming delocalisation perpendicular to the transport direction, that is to say a real $k_y$, then Dirac point transport is governed by the purely evanescent momenta $k_x = \pm i k_y$. For a single terrace geometry this results in the well-known minimal conductivity of bilayer graphene, a phenomena underpinned by a transmission function allowing, as shown in Fig. 1d, charge tunnelling through the device for only a very limited range of imaginary momenta (of the order of $1/L^2$). Furthermore, the window for which a terrace is open to charge transport depends significantly on its length, with the resonance centre given by

$$\text{Im } k_x^{(R)} = \frac{\ln L/l_\perp}{L} \tag{1}$$

where $l_\perp \approx 15$ Å. To bring this single terrace physics to the problem of a dislocation threaded sample, we simply insert an ideal (i.e., non-scattering) lead between the partial dislocation and the second terrace such that the transport problem is then recast as a multiple scattering problem involving single terrace transmission functions[19], with the total transmission given by $t = t_{AC}(1 - r_{AB}r_{AC})^{-1}t_{AB}$. In this expression $|t_{AB/AC}|^2$ are the transmission functions of the individual terraces, with $(1 - r_{AB}r_{AC})^{-1}$ encoding multiple scattering between them. From

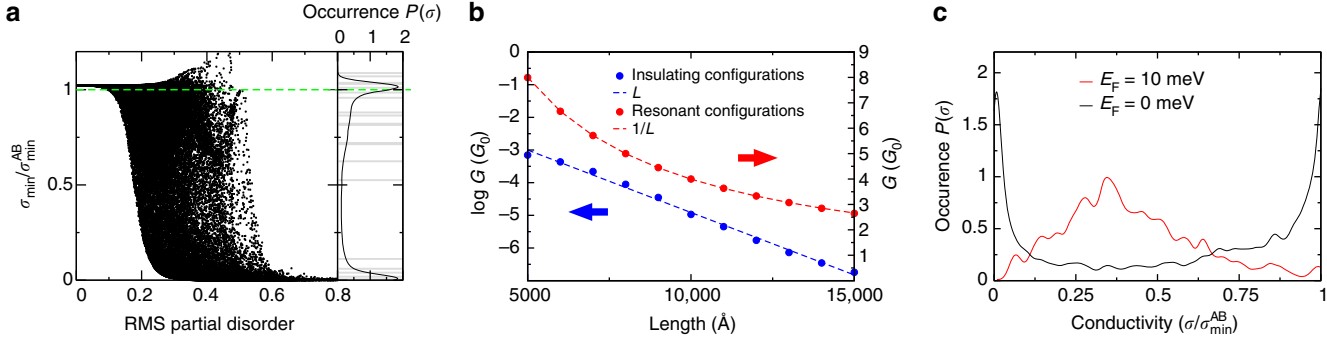

**Fig. 3** Insulating and metallic behaviour in evanescent transport through disordered configurations of partial dislocations. **a** For three partials, the minimal conductivity at the Dirac point is computed for an ensemble of randomly chosen partial positions, plotted vs. the RMS of the domain length distribution $L_{rms}^2 = \sum_{i=1,4} (L_i - \overline{L})^2 / \overline{L}^2$ of the partial configuration. Each *dot* represents one of 57,124 calculated realisations of the positions of three partial dislocations in a 1 μm sample. All transport states between resonant and fully insulating may be seen, however minimal metallic conductivity (indicated by the *dashed green line*) and fully insulating states are significantly more likely to be found than any other, as shown by the probability density function in the *right hand panel*. The *grey bars* represent experimental data[4], which show a qualitatively similar likelihood of occurrence of transport states (each *bar* represents a transport measurement in one suspended bilayer graphene sample). **b** Conductance as a function of sample length: for a restricted ensemble average over configurations close to minimal metallic conductivity (defined by $|\sigma_{min}/\sigma_{min}^{AB} - 1| < 10^{-3}$) a $L^{-1}$ dependence on length is seen, while for a restricted ensemble average performed over the insulating configurations ($|\sigma_{min}/\sigma_{min}^{AB}| < 10^{-3}$) the expected exponential decrease in conductivity with $L$ is found ($\sigma_{min}^{AB}$ represents the minimal metallic conductivity of bilayer graphene). In **c** the probability density function for conductivity at the Dirac point is shown and at a finite doping of 10 meV—only at the Dirac point is the two-state distribution function observed

this we see that transport will be profoundly influenced by the resonance centres $\mathrm{Im}\, k_x^{(R)}$ of the individual terraces, as a non-zero system transmission requires that $|t_{AB/AC}|^2$ have common evanescent momenta for which they are themselves non-zero. For a symmetric partial position, entailing AB and AC segments with similar $L$, $|t_{AB}|^2$ and $|t_{AC}|^2$ do have coinciding ranges of their open values of $\mathrm{Im}\,k_x$ and so a conducting state is found (Fig. 1e). On the other hand for an unsymmetrical partial position, i.e., terraces of different length, there is no overlap between their open values of $\mathrm{Im}\,k_x$, and so transport is blocked (Fig. 1f).

This physics may also be understood in an intuitive way from the point of view of the evanescent transport wavefunction. The dominant component of such an exponential wavefunction has the form $x/l_\perp e^{-\mathrm{Im}\,k_x x^2}$, a result that may be straightforwardly obtained by expansion of the standard bilayer eigenfunction with the assumption of vanishing $E$ and finite $k_y$. Precisely this form can be seen in, for example, the A1 component—the projection onto sublattice A of layer 1—of the single terrace wavefunction shown in panel (a) of Fig. 2. This wavefunction must be matched to a lead component $|\Psi_{lead}| < 1$ at $x = L$ giving us the expression $L/l_\perp e^{-\mathrm{Im}\,k_x L} = \Psi_{lead}$ which, for large $L$, can be recast as $\mathrm{Im}\,k_x = \ln(L/l_\perp)/L$. This is the matching condition between the right lead and the evanescent bilayer wavefunction and, as might be expected, is exactly equal to the resonance centre of the single terrace transmission functions, Eq. (1). As a similar argument holds for the matching at a partial dislocation, we see that the high symmetry partial position will generate consistent matching conditions as the two terraces are of similar length. Indeed, as may be seen the transport states in Fig. 2a, b have a very similar form. On the other hand for a low symmetry partial position $x_1/L = 0.75$—panel (c) of the same figure—the terraces are of different length and the matching condition cannot be satisfied with the same $\mathrm{Im}\,k_x$ in each terrace. As a consequence of these incompatible matching conditions, the wavefunction in the second terrace collapses, and a blocking of charge transport ensues. While two-state transport for a single partial configuration can thus be clearly motivated in terms of the interplay between evanescent transport and partial dislocations, it generates the expectation that only high symmetry (i.e., highly ordered)

configurations of partials will conduct. Interestingly, as we now show, this is not the case.

**Transport in the case of multiple partial dislocations**. For more than one partial, the generic case is disordered among the partial coordinates $\{x_i\}$ which, as each partial is defined by one coordinate, may be characterised by a single number—the RMS of the terrace length distribution $L_{rms}^2 = \sum_i (L_i - \overline{L})^2 / \overline{L}^2$. In Fig. 3a we consider the case of three partial dislocations, and plot an ensemble of 57,124 randomly generated configurations against $L_{rms}$. Three distinct regions of transport vs. $L_{rms}$ may be seen. For small $L_{rms} < 0.1$, corresponding to ordered configurations in which the partials are approximately uniformly spaced, the transport resembles that of the high symmetry single partial configuration: a few percent resonant enhancement over the minimal metallic conductivity of pristine bilayer graphene. This is the multi-partial geometry directly analogous to the central position in the case of a single partial, and conducts for a similar reason: each terrace shares similar values of evanescent momentum for which transport is allowed. Contrasting this, in the limit of large $L_{rms} > 0.65$ the transport is seen to be completely blocked. This $L_{rms}$ regime corresponds to case of a partial pairing geometry in which two of the three partials have come close to their minimal allowed separation of 125 Å. This is analogous to the unsymmetrical partial position, as the very different terrace lengths lead to non-coinciding transmission windows and thus to transport blocking. Most interesting, however, is the intermediate no-pairing but disordered regime. In this region the same value of $L_{rms}$ is found to yield almost all transport states from quite strong resonant enhancement ($\approx 20\%$ above the conductivity of structurally perfect bilayer graphene) to complete suppression of transport. The probability density function $P(\sigma)$ (Fig. 3a) again shows pronounced peaks at both minimal metallic conductivity and strong suppression of transport, just as in the case of a single partial dislocation. Two-state transport is thus robust against disorder among partial coordinates for multi-partial configurations—a very surprising result if one considers that in the case of a single partial the two-state transport was driven by a high symmetry resonant state. This two-state transport, it should

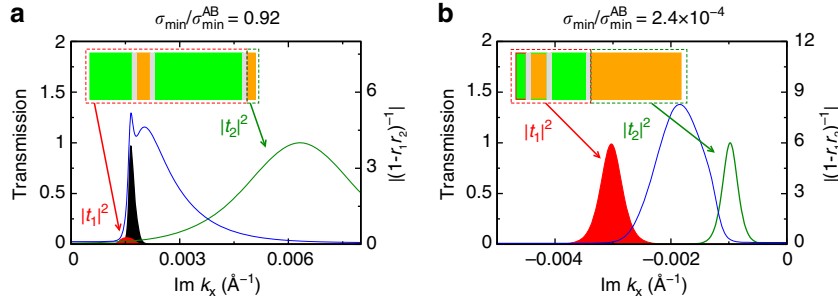

**Fig. 4** Multiple scattering at the boundary between bilayer terraces. Two partial configurations with similar terrace disorder $L_{rms} \approx 0.4$ exhibit two very different transport states: conducting **a** and insulating **b**. *Green/orange shading* indicates AB/AC stacked terraces, with *grey* denoting the partial dislocations (not drawn to scale). By division of the sample into two sub-systems, as indicated by the *dashed boxes*, the transport problem can be expressed in terms of the transmission functions of the sub-systems $t_{1/2}$ and multiple scattering between the sub-systems $(1 - r_1 r_2)^{-1} = 1 + r_1 r_2 + (r_1 r_2)^2 + ...$, with $|r_1|^2$ $(|r_2|^2)$ the probability of a left (right) moving wave reflecting from system 1 (2). The origin of the conducting state is then seen to lie in the coincidence of a sub-system transmission maximum with the maximum of the multiple scattering term—clearly present in system **a**, but not in system **b**—or, in other words, in a resonance between bilayer terraces

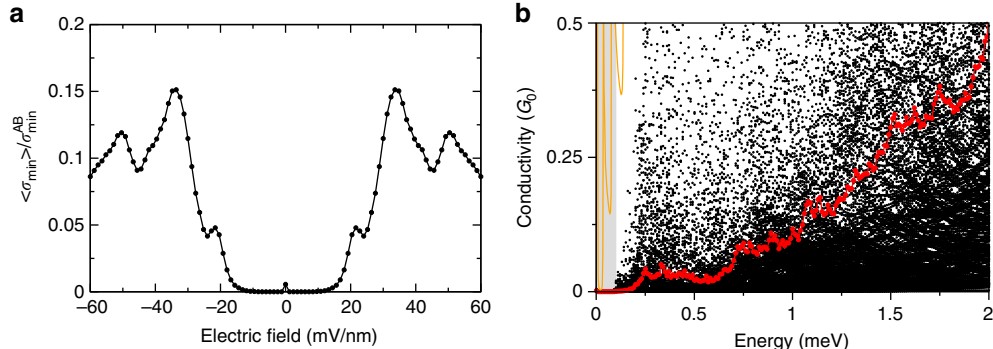

**Fig. 5** Restoration of the metallic state. Dependence of the partial dislocation induced insulating state on **a** layer symmetry breaking potential $V$ and **b** the Fermi energy $E_F$. In both panels the conductivities are obtained by averaging over a large ensemble of partial configurations that, for zero field and at charge neutrality, result in an insulating state (defined by $\sigma_{min}/\sigma_{min}^{AB} < 0.001$, where $\sigma_{min}^{AB}$ represents the minimal metallic conductivity of bilayer graphene). In **b** the individual points represent members of the ensemble of conductivities, while the *red line* denotes the ensemble average. The *orange line* in **b** is the conductivity of a perfect dislocation free bilayer, seen to be very different from the dislocation threaded ensemble average. The *grey shaded region* indicates the energy window within which transport is blocked for all partial configurations

be stressed, is a feature of transport at the Dirac point, and does not hold at any finite energy (see Fig. 3c for a comparison of $P(\sigma)$ at the Dirac point and at a finite energy of 10 meV). Finally, to make a direct comparison with experiment we take the data from ref. [4] which, after rescaling, is presented by the *grey bars* in the *right hand panel* of Fig. 3a. Each *bar* represents a transport measurement in a single suspended bilayer graphene sample, and the experimental distribution of conductivities is seen to match very well the theoretical probability density function.

To probe the reason behind the occurrence of insulating as well as conducting samples in the intermediate disorder regime, we perform a scattering analysis for two configurations having nearly identical $L_{rms} \approx 0.4$ but very different transport states. In a similar way to the analysis for the single partial configuration, we break each bilayer sample into two sub-systems: the first three terraces and partial dislocations and the last terrace (Fig. 4). The overall transmission is then obtained from multiple scattering between these sub-systems as $t = t_2 (1 - r_1 r_2)^{-1} t_1$ [19], where $|t|^2$ is the transmission probability through the whole system, $|t_{1/2}|^2$ the sub-system transmission probabilities and $|r_1|^2$ $(|r_2|^2)$ the probability of a left (right) moving wave reflecting from sub-system 1 (2). As may be seen in Fig. 4, although the maxima of the sub-system transmission functions are widely separated in both cases, leading on the basis of the single partial analysis to the

expectation of a blocked transport, for the conducting bilayer—system (a) in Fig. 4—the maxima of $|(1 - r_1 r_2)^{-1}|$ and $t_1$ coincide, resulting in a pronounced enhancement of the system transmission function $t$. Given that the term $(1 - r_1 r_2)^{-1} = 1 + r_1 r_2 + (r_1 r_2)^2 + ...$ encodes multiple back and forth scattering at the boundary between the two sub-systems, this enhancement corresponds physically to a multiple scattering resonance between bilayer terraces (as opposed to the evanescent resonance intrinsic to a single terrace). It is the presence of such resonances, absent in system (b) for which the maxima of $|(1 - r_1 r_2)^{-1}|$ evidently does not coincide with the maxima of either $t_{1/2}$, that is responsible for the existence of both insulating as well as conducting samples in the intermediate disorder regime.

To further connect with experiment we now investigate the physical properties of the insulating configurations seen in Fig. 3a. To this end we first define, from the full ensemble, an insulating ensemble consisting only of states that satisfy $|\sigma_{min}/\sigma_{min}^{AB}| < 10^{-3}$. We first confirm the insulating character of this reduced ensemble from the exponential form of the ensemble averaged $\langle \sigma(L) \rangle$ (Fig. 3b). Similarly, we may define a minimal metallic ensemble by the restriction $|\sigma_{min}/\sigma_{min}^{AB} - 1| < 10^{-3}$ which, reassuringly, displays the expected algebraic $L^{-1}$ length dependence characteristic of the minimal metallic state (Fig. 3b). We now consider the impact on the insulating ensemble of

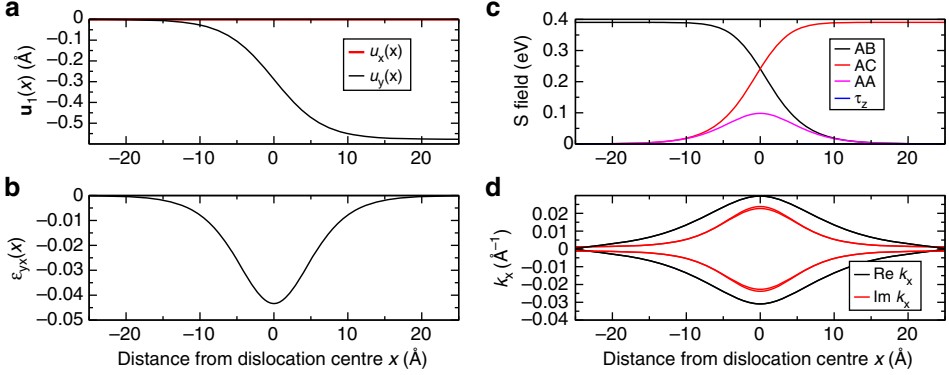

**Fig. 6** Continuum description of a partial dislocation. We show here the case of a type three partial; types 1 and 2 show similar behaviour. **a** The displacement vector of the two layers $\mathbf{u}(x)$, with the first layer is shifted by $\mathbf{u}(x)$ with respect to the second layer. **b** The strain state of the partial $\varepsilon_{xy}$ showing that the partial is under a modest strain taking its maximum value at the partial mid point. **c** The resulting effective field $S(x)$, see Eqs. (2) and (5), expressed as its decomposition onto a complete set of stacking matrices, AB, AC, AA and $\tau_z$ stacking, see Eq. (9). Notice that the $\tau_z$ field is identically zero, and is in fact zero for all partial geometries. **d** The transport momenta. While this is purely imaginary within the terraces it is seen to acquire a real part within the partial dislocation

introducing a layer perpendicular field $E_\perp$ and changing the particle number.

Application of a layer perpendicular field $E_\perp$ weakly restores the metallic state, see Fig. 5a, which in our model follows simply from the fact that an applied bias results in the well-known Mexican hat dispersion[3], thus imbuing the AB/AC segments with a small but finite momentum and so opening a propagating channel between them. As in experiment[20, 21], further increase in $E_\perp$ results once again in a suppression of the minimal metallic state. Increase in the bilayer density by top gating is also known to restore transport and, as may be seen in Fig. 5b, our model also captures this effect with, most interestingly, a gap of 0.2 meV —within which the insulating ensemble remains fully insulating for all members—of a similar magnitude (0.2 meV) to that reported in experiment[4]. This has a qualitatively similar origin as restoration by field: an increase in energy will allow real momenta in the transport direction (as one is no longer at the Dirac point) and so will reopen propagating channels in the transport.

In both Fig. 5a, b it will be noted that the reestablishment of the metallic state is rather weak, a fact which prompts a discussion of the straight partial model we have employed. The inherent translational symmetry of this model results in a separation of real lead momenta ($k_y$) from, at the Dirac point, imaginary transport momenta ($k_x$), which in an irregular partial geometry will mix. At the Dirac point, however, states of real momenta are apparently blocked by partials[8] and so we expect our Dirac point results to be robust to this more complex mode matching. On the other hand at finite doping the re-established propagating channel is expected to generate a higher conductivity in an irregular partial geometry than in a straight partial geometry, precisely due to this mixing of momenta. We thus expect the weaker establishment of the metallic state at finite doping and finite symmetry breaking field to be the most significant impact of the straight partial approximation.

## Discussion

We have shown that the approximately equal occurrence of both conducting and insulating transport states in experimental samples of bilayer graphene may be explained by an anomalous Dirac point transport involving partial dislocations and transport by evanescent states. This is underpinned by the fact that random arrangements of partial dislocations can support charge transport, leading to minimal metallic conductivity for arrangements of partials that one might expect to block transport. For an ensemble

of systems this leads to an unusual two-state distribution function of conductivities in which minimal metallic and insulating states occur with approximately equal probability at the Dirac point. The dramatically different transport states found in experiment are thus simply a manifestation of this hidden structural degree of freedom, which will naturally change from sample to sample as well as be changed by annealing the same sample. Finally, we note that ultra-clean suspended bilayer graphene has, in addition to interesting zero field transport physics, interesting physics at finite magnetic fields, in particular possible indications of a fractional quantum Hall effect[22–24]. In the light of the role that structural disorder plays in explaining the physics at zero field, it will be of interest to examine the impact of structural disorder at finite fields.

## Methods

**Overview**. For the transport calculation we use the Landauer formalism, appropriate as the experimental system is mesoscopic in dimensions, ultra-clean, and at low temperatures. Transport is thus likely to be well described by the phase coherent approach. In this context it is worth nothing the recent work of Bao et al.[4] in which two distinct types of bilayer graphene are reported: those having a very high mobility, for which two-state transport is observed, and those with a substantially lower mobility that were found always to be conducting. As the high mobility samples presumably exclude the kinds of single layer disorder likely to significantly disrupt the terrace wavefunction, we infer that bilayer terraces free from disorder are required for the two-state mechanism to hold.

In what follows we will describe the three distinct aspects of this calculation: obtaining the system Hamiltonian; determination of the transfer matrices of the bilayer sample; and finally deriving the overall scattering matrix and conductance from these transfer matrices.

**System Hamiltonian**. The multi-scale nature of the transport problem mandates the use of a continuum approach in which the dislocation threaded bilayer is treated via some generalisation of the well-known Dirac–Weyl equation of graphene. Recently, a general and exact scheme has been presented by which the tight-binding Hamiltonian of any system can be directly mapped to a effective continuum $H(\mathbf{r}, \mathbf{p})$ Hamiltonian[15], and we will employ this method here (see also ref. [11] where this method has been deployed to treat a partial dislocation network in bilayer graphene). For our tight-binding method we employ two hopping functions $t_\parallel(\delta) = A_\parallel e^{-B_\parallel \delta^2}$ and $t_\perp(\delta) = A_\perp e^{-B_\perp \delta^2}$ that describe in-plane and interlayer hopping, respectively, with the argument $\delta$ the length of the hopping vector. The in-plane constants $A_\parallel$ and $B_\parallel$ are obtained by optimising the band structure from a standard tight-binding calculation to reproduce a Fermi velocity $v_F = 10^6$ ms$^{-1}$ and the known in-plane nearest neighbour hopping strength of 2.8 eV, while the interlayer constants $A_\perp$ and $B_\perp$ are similarly fixed to reproduce the energy separation of the bonding and anti-bonding states in AB stacked bilayer graphene (0.78 eV) while maintaining a nearest neighbour hopping strength of 0.38 eV.

The key structural descriptor for the continuum approach is a stacking field $\mathbf{u}(x)$ that describes the local relative displacement of the two layers of bilayer

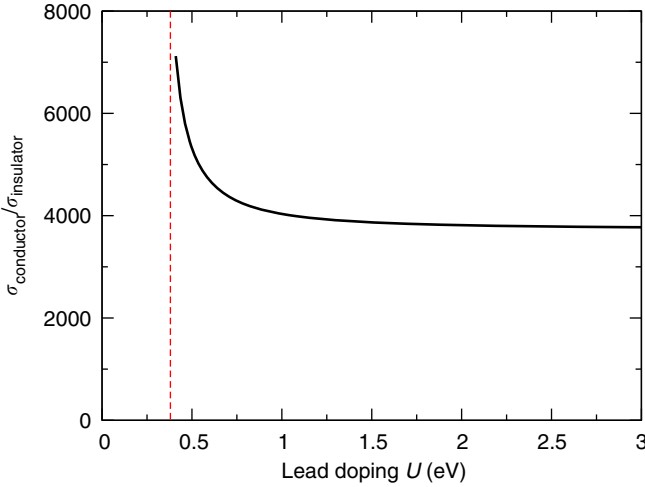

**Fig. 7** Dependence of two-state transport on lead doping. Shown are the ratio of the conductivities of the insulating and conducting systems exhibited in Fig. 4 **a**, **b** as a function of the lead doping $U$. Evidently the dramatically different transport states of the two bilayer samples are not significantly impacted by the lead doping

graphene from some initial reference stacking due to, for example, a series of partial dislocations. For the case of a single type, three partial dislocation such a stacking field is shown in Fig. 6a. Without loss of generality we may take the second of the two layers of the bilayer to be fixed, and apply the shift field $\mathbf{u}(x)$ only to the first layer. In this case the effective Hamiltonian for a graphene bilayer system with a general $\mathbf{u}(x)$ field is given by

$$H = \begin{pmatrix} \hbar v_F \boldsymbol{\sigma}.(\mathbf{p} + \mathbf{A}(x)) + \mathbb{1}_2 V(x) & S(x) \\ S^\dagger(x) & \hbar v_F \boldsymbol{\sigma}^*.\mathbf{p} \end{pmatrix} \quad (2)$$

where $\mathbb{1}_2$ is the $2 \times 2$ unit matrix, and $v_F$ the Fermi velocity of graphene. The diagonal blocks of this Hamiltonian are the Dirac–Weyl operators for each layer, with the first layer operator augmented by fields arising from strain due to the application the displacement field $\mathbf{u}(x)$ to this layer, while the off diagonal blocks encode the complex spatially dependent interlayer stacking. The strain induced layer–diagonal fields are given by the standard results[25] with the effective scalar field given by

$$V(x) = C_V u_{xx} \quad (3)$$

and effective pseudo-gauge field by

$$\mathbf{A}(x) = C_A(-u_{xx}, u_{yx}), \quad (4)$$

where the constants $C_V$ and $C_A$ are the coupling strengths (that depend on the in-plane hopping constants $A_\parallel$ and $B_\parallel$, see ref. [15]) and $u_{ij} = \partial_j u_i$ the components of the strain tensor with $u_i$ the $i$th component of the displacement field $\mathbf{u}(x)$. The strain associated with a type three partial is shown in Fig. 6b. The interlayer effective field is a considerably more complex object and is given by refs. [11, 15]

$$S(x) = \frac{1}{A_{UC}} \sum_i M_i t_\perp(K_i) e^{-iK_i.\mathbf{u}(x)}. \quad (5)$$

where $A_{UC}$ is area of the bilayer unit cell. Note that this field—and all subsequent objects related to it—is a $2 \times 2$ matrices in interlayer sublattice space, i.e., each element represents an electron hopping amplitude between the sublattices of different layers. In this expression the sum $i$ is over the translation group of the high symmetry point $\mathbf{K} = 2\pi/a(2/3,0)$: $\mathbf{K}_i = \mathbf{K} + \mathbf{G}_i$ where $\mathbf{G}_i$ are the reciprocal space lattice vectors of bilayer graphene. The matrices $M_i$ are given by $[M_i]_{\alpha\beta} = e^{i\mathbf{G}_i.(\boldsymbol{\nu}_\alpha - \boldsymbol{\nu}_\beta)}$ where $\boldsymbol{\nu}_\alpha$ ($\boldsymbol{\nu}_\beta$) the basis vectors of layer one (two) of the bilayer. Finally, the function $t_\perp(q)$ is the Fourier transform of the interlayer hopping function $t_\perp(\delta)$ where $\delta$ is the separation vector of the two sites (in different layers) that the electron hops between: $\delta = |\mathbf{R}_j - \mathbf{R}_i|$. As this function, and hence its Fourier transform $t_\perp(q)$, are exponentially decaying a first star approximation in which the sum is restricted to include only $\mathbf{K}$, and the two equivalent high symmetry $K$ points of the bilayer Brillouin zone already constitutes a good approximation for $S(x)$. Assuming the AB structure as a reference state, and taking a standard choice of basis vectors $\boldsymbol{\nu}_1 = \mathbf{0}$ and $\boldsymbol{\nu}_2 = a(1, 1/\sqrt{3})$, the three $M_i$ matrices

in this case are found to be

$$M_1 = \begin{pmatrix} 1 & 1 \\ & \\ 1 & 1 \end{pmatrix}, \quad M_2 = \begin{pmatrix} 1 & e^{-i2\pi/3} \\ & \\ e^{-i2\pi/3} & e^{i2\pi/3} \end{pmatrix}, \quad M_3 = \begin{pmatrix} 1 & e^{i2\pi/3} \\ & \\ e^{i2\pi/3} & e^{-i2\pi/3} \end{pmatrix} \quad (6)$$

The crucial aspect of this field $S(x)$ is that it treats AB, AC, and any intermediate stacking on exactly the same footing. To see this note that insertion of $\mathbf{u} = \mathbf{0}$ into Eq. (5) yields, as it must, the interlayer coupling of the reference state AB bilayer

$$S(x) \rightarrow S_{AB} = \frac{t_K}{A_{UC}} \begin{pmatrix} 1 & 0 \\ & \\ 0 & 0 \end{pmatrix}, \quad (7)$$

while, on the other hand inserting any of the three nearest neighbour vectors of graphene, for instance $\mathbf{u} = a(0, -1/\sqrt{3})$, gives exactly the interlayer coupling of the AC bilayer:

$$S(x) \rightarrow S_{AC} = \frac{t_K}{A_{UC}} \begin{pmatrix} 0 & 0 \\ & \\ 0 & 1 \end{pmatrix} \quad (8)$$

The partial dislocation, which interpolates between these two limits, we model using the displacement field $\mathbf{u}(x)$ shown in panel of Fig. 6 for the case of a partial Burgers vector $a(0, -1/\sqrt{3})$. This form of $\mathbf{u}(x)$ is chosen to reproduce as closely as possible the calculated partial structure described in ref. [10]. As may be seen—see panel (b) of Fig. 6—the partial dislocation is strained, with the maximum strain found at the partial core. The resulting interlayer coupling field $S(x)$ can be conveniently expressed by projection onto a complete set of four stacking matrices which, with a $c$-number coefficient, are sufficient to describe all possible interlayer coupling fields: $S(x) = c_{AB}(x)\tau_{AB} + c_{AC}(x)\tau_{AC} + c_{AA}(x)\tau_{AA} + c_z(x)\tau_z$. These matrices are given by

$$\tau_{AB} = \begin{pmatrix} 1 & 0 \\ & \\ 0 & 0 \end{pmatrix}, \quad \tau_{AC} = \begin{pmatrix} 0 & 0 \\ & \\ 0 & 1 \end{pmatrix}, \quad \tau_{AA} = \begin{pmatrix} 0 & 1 \\ & \\ 1 & 0 \end{pmatrix}, \quad \tau_z = \begin{pmatrix} 0 & 1 \\ & \\ -1 & 0 \end{pmatrix} \quad (9)$$

This projection, as the $\tau$ matrices evidently inhabit the interlayer and sublattice space of the $S(x)$ field, is directly informative of the local stacking of the bilayer. This projection is exhibited in panel (c) of Fig. 6 revealing the expected transition from AB to AC stacking across the partial but also a non-zero AA component of the stacking field, that has its maximum at the partial core. The projection onto the $\tau_z$ type of stacking is zero throughout the partial dislocation.

**Sample transfer matrices.** Having established the system Hamiltonian $H(x)$, we proceed by dividing the bilayer sample into a series of small strips (we use a width of 100 Å for the terraces and 0.2 Å for the partial dislocation) within which $H(x)$ can be considered constant. By expressing the strip Hamiltonian as $H = H_0 + H_x k_x + H_y k_y$ we may rewrite the eigenvalue problem $H\Psi = E\Psi$ as

$$H_x^{-1}(\mathbb{1}_4 E - H_0 - H_y k_y)\Psi = k_x \Psi \quad (10)$$

($\mathbb{1}_4$ is the $4 \times 4$ unit matrix), which yields four $k_x^{(i)}$ eigenvalues for given values of $E$ and $k_y$, the constants of motion of the transport problem (translational symmetry is assumed in the $y$ direction). These $k_x^{(i)}$ are the transport momenta, and are shown in panel (d) of Fig. 6 plotted across the same type three partial dislocation for which the displacement field $\mathbf{u}(x)$ is presented in panel (a). From these, and the corresponding eigenfunctions $\Psi_i$, we can easily construct the transfer matrix of the strip. Denoting as $W$ that matrix in which each column is one of the eigenfunctions $\Psi_i$, and as $B$ the diagonal matrix with elements $e^{ik_x^{(i)}\Delta L}$ where $\Delta L$ the strip width, then the transfer matrix is simply $WBW^{-1}$.

**Transport calculation.** Following the usual approach adopted in calculations of Dirac point conductivity, we model the leads as highly doped bilayer graphene, see ref. [2]. The leads are encoded in a $W_L$ matrix in which each column consists of a highly doped bilayer graphene wavefunction (two right moving and two left moving states), for an explicit form of $W_L$ see the appendix of ref. [2]. We have checked that the qualitative nature of the insulating and conducting partial

dislocation configurations is independent of the doping of the lead (Fig. 7). As may be seen, deviation from the limiting case of high doping is found only close to the edge of the bilayer anti-bonding band.

The remainder of the calculation follows the standard methodology of the scattering theory Landauer approach, see refs. [2, 19] and Snyman. If the sample consisted of only one strip then the overall system transfer matrix would be

$$M_{\text{system}} = W_{\text{L}}^{-1} W B W^{-1} W_{\text{L}}, \tag{11}$$

from which the system scattering matrix $S_{\text{system}}$ could then be derived. However, as the system consists of many such strips this is not the case. Instead, insertion of leads between each strip (which does not change the transport properties as these do not scatter) yields a series of strip transfer matrices $M_{\text{strip}}$ having exactly the form of Eq. (11), from which a corresponding series of strip scattering matrices can be derived. These are then, by iterative use of the equations for combining two scattering matrices[19], combined to give the overall system scattering matrix $S_{\text{system}}$. From this the conductance $G$ is then obtained via the Landauer formula.

**Data availability**. The data that support the findings of this study are available from the corresponding author upon request.

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

## Acknowledgements

The work was carried out in the framework of the SFB 953 and SPP 1459 of the Deutsche Forschungsgemeinschaft (DFG).

## Author contributions

The project was initiated by H.B.W. and S. Shallcross. The theoretical development, code development and calculations were performed by SShall. The analysis of results and writing of the manuscript were the joint enterprise of S. Shallcross, H.B.W. and S. Sharma.

## Additional information

**Competing interests:** The authors declare no competing financial interests.

