## [Peer Review File · Nature Communications]

Reviewers' Comments:

Reviewer #1 (Remarks to the Author)

The manuscript submitted as a publication in Nature Communications "Anomalous Dirac point transport due to extended defects in bilayer graphene" addresses a well-known and definitely important problem in the topic of the electronic transport regime of ultra-clean samples of bilayer graphene, of which approximately half exhibit metallic behavior, otherwise insulating. Due to the scarcity of other sources of scatterers such as phonons, other electrons, extrinsic defects one is indeed led to ascribe the emergence of insulating behavior due to the presence of stacking faults (also called partials). The authors suggest that the spatial distribution of the defects determine the transport behavior. However, the detailed mechanisms proposed by the authors is not convincing to me (at least in the present form of the manuscript). Possibly, this is due to the fact that the I do not fully understand the method and models employed. In particular:

1. What are the boundary conditions used, besides the generic definition "hard" (which, I assume, are $U(0, L)=\infty$) and "soft" leads?

2. The model Hamiltonian is briefly sketched in the Methods section. As a suggestion, either one fully explains the method -- I recall there is no length constraint on this part -- or fully reference other sources. The method section of this manuscript is just useless, if not frustrating for the reader. For example, in which space do the tau matrices operate (I assume sub-lattice space)? What is the relative value of the interlayer vs intralayer coupling? How do you exactly couple the leads to the scattering region in the Landauer approach (especially in case of "hard leads" how are oscillating wave-functions injected into the sample)?

3 (Minor point). It is not true that diffusive transport cannot be treated within Landauer approach. This is indeed a frequently misunderstood point. What cannot be treated (in the original Landauer approach) is incoherent transport, namely when the motion of the electrons is not phase-conserving because of lattice dynamics, for example. Impurity scattering, treatable within Landauer, can well result in diffusive transport ($\sigma \propto 1/L$).

Concerning the results:

1- Case of the individual defect: the conductivity approaches the minimal universal value if the dislocation is in high symmetry position, i.e. at the center of the sample. This behavior is expected if one assume closed boundary conditions ("hard leads"). It is no surprise that the authors find a similar behavior in the case of exponential behavior of the wave-function outside the leads. This is obviously not true if one assumes infinite leads of pristine bilayer graphene, for it would not make sense to speak of a high-symmetry position of the partial. Of course, this is an unreal limiting case, but makes clear how the leads can have a substantial effect on the model. The typical model situation for making minimum universal conductivity emerge is that one assumes doped bilayer graphene for the leads, and intrinsic graphene for the scattering region (ex. Ref 1. and 2.). Why did not the authors use the same approach here, which also gives the possibility of tuning the level of doping in the leads?

2. The manuscript suggest that the presence of parallel partials implies that electrons are confined in a series of quantum wells delimited by the position of the partials. I understand that imaginary value of k_x is key to prevent localization due to destructive interference, but why in certain cases - the most symmetric ones -- electrons can effectively tunnel from one well to the next, and in other case -- the less symmetric -- not? A necessary condition to re-evaluate my position is that this aspect is adequately discussed and convincingly clarified.

3. The author, speculate that the single model category that they treat, namely partials along armchair direction and perpendicular to the transport direction, does not hinder the generality of the result. I wonder how they would extend the disorder quantifier in the case of non parallel partials, which is indeed very common from experimental results.

Concerning the figures:

1. Insets of figure 2, given the units (\AA^{-1}) and the absence of reference to the 1D projected Brillouin zone are just not readable. I guess non-zero transmission only occurs in the vicinity of $k_y=2\pi/3a$, a being the periodicity along y axis. Still, what is the information that the reader should take from those plots?

To be honest, it has to be said that the idea is intriguing and the resemblance to the experimental results is satisfying. However, this is not per se a sufficient reason to accept it as the definitive explanation of the important problem addressed.

For these reasons, I cannot recommend publication in Nature Communication, but I am open to reconsider the manuscript if the points that I have raised will be fully met, if possible by means of the additional calculations and explanations suggested.

Fernando Gargiulo

Reviewer #2 (Remarks to the Author)

The Authors carry out numerical simulations with a model Hamiltonian to get insights into the electronic transport at the Dirac point in bilayer graphene with extended defects. They first analyze the transport through a single partial dislocation and then study that in the essentially macroscopically large system with a random network of partial dislocations. The Authors reveal the origin of the experimentally observed insulating and metallic transport states in bilayer graphene, which was not fully understood.

Taking into account the large amount of attention currently being paid to electronic properties of graphene and other 2D materials, the results reported in the manuscript should be of interest to the readership of Nature Communications, and their presentation is good. The manuscript is well written, so that I recommend it for publication provided that the Authors consider discussing the following issues:

1. As the readership of Nature Communications includes a broad range of materials scientists, chemists and physicists, perhaps the Authors could explain in more detail the physical origin of the dislocations they consider (strain, rippling, etc.) and emphasize that they are fundamentally different from the in-plane dislocations in graphene [see, e.g., O.Yazyev's works, Nat. Nanotech. 9 (2014) 755 and references therein], the picture that most readers will likely have in mind when dislocations in graphene are mentioned.

2. Following from the above, the Authors may also discuss the relationship between point defects, in plane (e.g., edge) dislocations and partial dislocations they consider. As the experiments [Nature Communications 4 (2013) 2098] indicate, in graphene splitting of point defects (e.g., divacancies) due to carbon bond rotations can be equivalent to the formation of several dislocations, which in turn give rise to buckling and in principle, may result in a change in the local stacking of the two sheets in the bilayer. Will the suggested model work if any disorder is present in the graphene sheets? If not, and if the two transport states would be completely smeared out, this could be another indirect confirmation for amazingly low concentration of defects in graphene.

3. Some technical issues: Font size should be increased in all the figures. Axes captions are barely readable, especially in the insets.

1 Reply to Referee 1

The manuscript submitted as a publication in Nature Communications "Anomalous Dirac point transport due to extended defects in bilayer graphene" addresses a well-known and definitely important problem in the topic of the electronic transport regime of ultra-clean samples of bilayer graphene, of which approximately half exhibit metallic behaviour, otherwise insulating. Due to the scarcity of other sources of scatterers such as phonons, other electrons, extrinsic defects one is indeed led to ascribe the emergence of insulating behavior due to the presence of stacking faults (also called partials). The authors suggest that the spatial distribution of the defects determine the transport behavior. However, the detailed mechanisms proposed by the authors is not convincing to me (at least in the present form of the manuscript). Possibly, this is due to the fact that the I do not fully understand the method and models employed. In particular:

Concerning the model:

1. What are the boundary conditions used, besides the generic definition "hard" (which, I assume, are $U(0, L) = \infty$) and "soft" leads?

This is a misunderstanding caused by our eccentric choice of notation (and a typo – "exponential envelope") in the previous version: our lead/material formalism is in fact exactly that used in previous calculations of minimal conductivity, i.e., doped bilayer graphene for the leads and intrinsic bilayer graphene for the material. The notation "soft" implied a finite lead doping, while the notation "hard" the limit when this doping goes to infinity.

2. The model Hamiltonian is briefly sketched in the Methods section. As a suggestion, either one fully explains the method – I recall there is no length constraint on this part – or fully reference other sources. The method section of this manuscript is just useless, if not frustrating for the reader. For example, in which space do the τ matrices operate (I assume sub-lattice space)? What is the relative value of the interlayer vs intralayer coupling? How do you exactly couple the leads to the scattering region in the Landauer approach (especially in case of "hard leads" how are oscillating wave-functions injected into the sample)?

We have both substantially extended the Methods section as well as incorporated additional references into it. We hope it now offers a clear background guide to the calculations performed in the manuscript. In particular, (i) the construction of the Hamiltonian is now explained in detail and this should make objects like the τ matrices much clearer to understand (which are indeed in sub-lattice space as the Referee correctly points out), (ii) the underlying tight-binding method from which our continuum Hamiltonian is derived is described (including values of intra- vs. interlayer hopping constants), and (iii) the coupling of the leads to the device is now both clearly elucidated as well as clearly sourced (we follow closely the scheme of Refs. 1 and 2). Note that Ref. 15, which gathers together the theoretical methods used for construction of the model Hamiltonian, is still refereed to for theoretical background. As this is currently under review at PRX we cannot give a more substantial reference, although the method we employ has previously been used successfully to describe both the twist bilayer

46 [*Phys. Rev. B* 93, 035452 (2016)] and, perhaps more directly relevant, partial
47 dislocation networks in bilayer graphene [*Nat Phys* 11, 650653 (2015)].)

48 3 (Minor point). It is not true that diffusive transport cannot be treated
49 within Landauer approach. This is indeed a frequently misunderstood point.
50 What cannot be treated (in the original Landauer approach) is incoherent trans-
51 port, namely when the motion of the electrons is not phase-conserving because of
52 lattice dynamics, for example. Impurity scattering, treatable within Landauer,
53 can well result in diffusive transport ($\sigma \propto 1/L$).

54 *Our intention here was just to contrast typical large area “dirty” graphene*
55 *(diffusive and incoherent transport) with the ultra-clean samples of the exper-*
56 *iment under consideration. However, the Referee is correct to point out that*
57 *our statement gave the impression that the coherent approach could not treat*
58 *the case of impurity scattering leading to a $1/L$ diffusive transport. We have*
59 *accordingly rephrased this point, see lines 183-184 of the manuscript, and thank*
60 *the Referee for pointing this out.*

61 Concerning the results:

62 1. Case of the individual defect: the conductivity approaches the minimal
63 universal value if the dislocation is in high symmetry position, i.e. at the center
64 of the sample. This behavior is expected if one assume closed boundary condi-
65 tions (“hard leads”). It is no surprise that the authors find a similar behavior in
66 the case of exponential behavior of the wave-function outside the leads. This is
67 obviously not true if one assumes infinite leads of pristine bilayer graphene, for
68 it would not make sense to speak of a high-symmetry position of the partial. Of
69 course, this is an unreal limiting case, but makes clear how the leads can have
70 a substantial effect on the model. The typical model situation for making mini-
71 mum universal conductivity emerge is that one assumes doped bilayer graphene
72 for the leads, and intrinsic graphene for the scattering region (ex. Ref 1. and
73 2.). Why did not the authors use the same approach here, which also gives the
74 possibility of tuning the level of doping in the leads?

75 *As stated above, our modeling of the leads/device is very close to that of*
76 *Refs. 1 and 2, i.e. we use doped bilayer graphene for the leads and intrinsic (but*
77 *defected) bilayer graphene for the scattering region. Following the suggestion of*
78 *the Referee we have, in our extended Methods section, now included a figure*
79 *plotting the ratio of conductivities for two example insulating and conducting*
80 *partial configurations as a function of lead doping (Fig. 7). As may be seen, it*
81 *is only as the lead doping approaches the anti-bonding band edge, at which the*
82 *lead description breaks down, that a quantitative change in this ratio is seen.*
83 *Dirac point two state transport is thus robust to the value of the lead doping.*

84 2. The manuscript suggest that the presence of parallel partials implies that
85 electrons are confined in a series of quantum wells delimited by the position
86 of the partials. I understand that imaginary value of k_x is key to prevent
87 localization due to destructive interference, but why in certain cases – the most
88 symmetric ones – electrons can effectively tunnel from one well to the next, and
89 in other case – the less symmetric – not? A necessary condition to re-evaluate

my position is that this aspect is adequately discussed and convincingly clarified.

This is an important question, and has prompted us to search for a more intuitive explanation of our finding of two state transport at the Dirac point. This is now explained in a new paragraph, lines 74-90 of the manuscript as well as three additional panels to Fig. 2. In essence, each terrace is endowed with a transmission resonance allowing charge transport only for a limited range of evanescent momenta $Im k_x$, with the resonance centre $Im k_x^{(R)} = (\ln L/l_{\perp})/L$ and width ($\approx 1/L$) strongly dependent on the terrace length (see Fig. 1d). By expressing the overall transport, via a standard formula for combining scattering matrices, in terms of the transmission functions of the individual terraces, we find an intuitive explanation in terms of single terrace resonances. For terraces of similar length, that occur when the partial is at a high symmetry position, the resonance centers have similar $Im k_x$, and thus both terraces have common values $Im k_x$ at which they are “open” to charge transport. Overall the system thus conducts (see Fig. 1e). On the other hand for a low symmetry position of the partial, the different terrace lengths lead to substantially different centers of the evanescent resonances $Im k_x^{(R)}$, to no common values of evanescent momenta for which both terraces are open, and so to a blocking of charge transport (see Fig. 1f).

This physics can also be understood in a rather intuitive way in terms of the transport wavefunctions which indeed, as the Referee points out, give the appearance of quantum well states (even though they are in fact the wavefunctions for open and not quantum well boundary conditions). This a complementary point of view to the multiple scattering analysis that underpins the transmission resonance picture, and is based on the concept of consistent or incompatible matching conditions between the terrace wavefunctions and partial dislocations or leads. This is explained in detail in a second paragraph, lines 91-106.

We have also, using a similar multiple scattering analysis to that deployed for the single partial case, addressed the question of why for the intermediate disorder case either an insulating or conducting state can result from similarly disordered terrace geometries. We find that multiple scattering at the boundaries between terraces can lead to a resonant enhancement of transport, and demonstrate this in Fig. 3 for two cases of intermediate disorder, along with accompanying text in lines 131-145.

This more intuitive explanation of two state transport – basically hewing closer to the language of transport – has, we believe, considerably improved the manuscript and we are grateful to the Referee for raising this point.

3. The author, speculate that the single model category that they treat, namely partials along armchair direction and perpendicular to the transport direction, does not hinder the generality of the result. I wonder how they would extend the disorder quantifier in the case of non parallel partials, which is indeed very common from experimental results.

Non-straight and non-parallel partials, in addition to other structural complications such as possible AA segments and trilayer samples, will be treated in a future work. For samples of non-straight partials the classification of disorder

136 obviously becomes more involved (one of the motivating reasons for our effective
 137 1d model) however we envisage a likely classification strategy to involve the ad-
 138 dition two parameters to the descriptor of the partial: (i) the difference between
 139 the left and right edge partial positions $|x_L - x_R|$ and (ii) a “stiffness constant”
 140 of the partial guiding a random walk between x_L and x_R . We suspect, however,
 141 and preliminary results indicate, that the qualitative nature of our findings will
 142 not be changed in such a situation. Our expectation would rather be that it will
 143 lead to a smearing of the evanescent resonances, as for a terrace bounded by
 144 non-straight partials an effective $\langle L \rangle$ average is performed. As may be seen
 145 from panel (d) of Fig. 1 of the manuscript, provided $|x_L - x_R|$ is not compa-
 146 rable to the sample length then this smearing will not alter fundamental fact
 147 that there will exist two distinct types of partial geometry: (i) terraces of simi-
 148 lar average length that possess transmission functions “open” for similar $\text{Im } k_x$
 149 values (conducting samples) and (ii) terraces with very different average lengths
 150 that have transmission functions having no coinciding $\text{Im } k_x$ values (insulating
 151 samples). We therefore expect two state transport to be robust to a wandering
 152 partial geometry.

153 Concerning the figures:

154 1. Insets of figure 2, given the units (Angstrom⁻¹) and the absence of refer-
 155 ence to the 1D projected Brillouin zone are just not readable. I guess non-zero
 156 transmission only occurs in the vicinity of $k_y = 2\pi/3a$, a being the periodicity
 157 along y axis. Still, what is the information that the reader should take from
 158 those plots?

159 *The insets for these figures showed the corresponding transmission function,*
 160 *intended primarily to convey whether the partial geometry resulted in an insu-*
 161 *lating or conducting state. As these inset panels were very unclear we have*
 162 *now instead simply written the corresponding conductivity over each panel. The*
 163 *region of non-zero transmission depends in a complex way on the partial con-*
 164 *figuration; for a single terrace the centre of the evanescent resonance peak is at*
 165 *$(\ln L/l_\perp)/L$.*

166 To be honest, it has to be said that the idea is intriguing and the resem-
 167 blance to the experimental results is satisfying. However, this is not per se
 168 a sufficient reason to accept it as the definitive explanation of the important
 169 problem addressed.

170 For these reasons, I cannot recommend publication in Nature Communi-
 171 cation, but I am open to reconsider the manuscript if the points that I have
 172 raised will be fully met, if possible by means of the additional calculations and
 173 explanations suggested.

174 *We have performed the additional calculations suggested by the Referee – the*
 175 *impact of the doping value of the leads, and elucidation of why a high (low) sym-*
 176 *metry partial generates a conducting (insulating) state. Accompanied by changes*
 177 *to our explanations of the model and methodology, along the lines suggested by*
 178 *the review, we hope we have clarified the paper to the satisfaction of the Referee.*

2 Reply to Referee 2

179

The Authors carry out numerical simulations with a model Hamiltonian to get insights into the electronic transport at the Dirac point in bilayer graphene with extended defects. They first analyze the transport through a single partial dislocation and then study that in the essentially macroscopically large system with a random network of partial dislocations. The Authors reveal the origin of the experimentally observed insulating and metallic transport states in bilayer graphene, which was not fully understood.

180
181
182
183
184
185
186

Taking into account the large amount of attention currently being paid to electronic properties of graphene and other 2D materials, the results reported in the manuscript should be of interest to the readership of Nature Communications, and their presentation is good. The manuscript is well written, so that I recommend it for publication provided that the Authors consider discussing the following issues:

187
188
189
190
191
192

1. As the readership of Nature Communications includes a broad range of materials scientists, chemists and physicists, perhaps the Authors could explain in more detail the physical origin of the dislocations they consider (strain, rippling, etc.) and emphasize that they are fundamentally different from the in-plane dislocations in graphene [see, e.g., O. Yazyevs works, Nat. Nanotech. 9 (2014) 755 and references therein], the picture that most readers will likely have in mind when dislocations in graphene are mentioned.

193
194
195
196
197
198
199

This is an important point and we have added such a discussion, and the references suggested by the Referee, to the introductory part of the manuscript, see lines 37-42. We are grateful to the Referee for the suggestion to place our particular type of structural disorder in a broader context.

200
201
202
203

We would expect that most significant difference between stacking fault partial dislocations and single layer dislocations resides in the fact that the former can exist between structurally almost perfect graphene sheets, as only local strain is required for their existence, and not the significant changes in local bonding involved in single layer dislocations. As a consequence they are certainly to be expected for growth situations in which a natural strain is imposed, for example by a substrate, and indeed this is the case for bilayer graphene grown epitaxially on SiC in which a dense network of dislocations is formed.

204
205
206
207
208
209
210
211

2. Following from the above, the Authors may also discuss the relationship between point defects, in plane (e.g., edge) dislocations and partial dislocations they consider. As the experiments [Nature Communications 4 (2013) 2098] indicate, in graphene splitting of point defects (e.g., divacancies) due to carbon bond rotations can be equivalent to the formation of several dislocations, which in turn give rise to buckling and in principle, may result in a change in the local stacking of the two sheets in the bilayer. Will the suggested model work if any disorder is present in the graphene sheets? If not, and if the two transport states would be completely smeared out, this could be another indirect confirmation for amazingly low concentration of defects in graphene.

212
213
214
215
216
217
218
219
220
221

We believe that this is the case. In this respect it is worth noting the work of Bao et al. – Ref. 4 of the manuscript – in which two distinct types of bilayer samples are reported: those having a very high mobility, for which two state

222
223
224

225 *transport is observed, and those with a substantially lower mobility, found always*
226 *to be conducting. As the high mobility samples presumably exclude kinds of single*
227 *layer disorder likely to substantially disrupt the terrace wavefunction, we would*
228 *thus infer that terraces free from disorder are required for the mechanism we*
229 *describe to hold.*

230 *In other words, the scenario we advocate requires two graphene sheets that*
231 *have a very low concentration of in-plane defects, but nevertheless contain at*
232 *least one stacking fault in the form of a partial dislocation. We believe this is a*
233 *realistic situation, as in-plane defects are expected to have a higher energy cost*
234 *than interlayer stacking faults, reflecting the corresponding strong in-plane and*
235 *weak interlayer coupling.*

236 *This is an interesting point and we have added it to the manuscript in the*
237 *context of the related topic of the applicability of the methods of coherent trans-*
238 *port, see lines 182-188 of the new Methods section.*

239 **3.** Some technical issues: Font size should be increased in all the figures.
240 Axes captions are barely readable, especially in the insets.

241 *Font sizes have been increased in all Figures, which should now be readable.*

Reviewers' Comments:

Reviewer #1:

Remarks to the Author:

I recognize that the authors took my remarks seriously and addressed my concerns almost completely.

The only remark which they've left open is the one concerning generalization to their results to non-parallel partials. Their argument is that addressing this issue deserves a publication on its own, on which they are currently working.

I take this statement as a gentlemen agreement that the authors will commit on publishing the result of this further piece of research, being they in agreement or in contradiction to the present claims.

I do recommend publication of this piece of research on Nature Communication.

Reviewer #2:

Remarks to the Author:

I am fully satisfied with the response of the Authors to my comments and questions. The manuscript has improved, and in my opinion it merits now the high Nature Communications standards of novelty and impact, so that I recommend it for publication in its present form.

Reviewer #1 (Remarks to the Author):

I recognize that the authors took my remarks seriously and addressed my concerns almost completely.

The only remark which they've left open is the one concerning generalization to their results to non-parallel partials. Their argument is that addressing this issue deserves a publication on its own, on which they are currently working.

I take this statement as a gentlemen agreement that the authors will commit on publishing the result of this further piece of research, being they in agreement or in contradiction to the present claims.

I do recommend publication of this piece of research on Nature Communication.

Response: The authors accept fully the terms of the proposed "gentlemens" agreement.

Reviewer #2 (Remarks to the Author):

I am fully satisfied with the response of the Authors to my comments and questions. The manuscript has improved, and in my opinion it merits now the high Nature Communications standards of novelty and impact, so that I recommend it for publication in its present form.

Sincerely,

S. Shallcross, S. Sharma, and H. W. Weber